# Spiking Hybrid Attentive Mechanism with Decoupled Layer Normalization for Joint Sound Localization and Classification

## Abstract

Localizing and identifying sound sources simultaneously through binaural cues is a crucial ability of humans, which facilitates our perception of complex surrounding scenes. Brain-inspired Spiking Neural Network (SNN) offers an energy-efficient and event-driven paradigm thus it is highly suitable for simulating the signal processing of such perceptions in organisms. Despite recent progress, most existing approaches in SNNs solely focus on a single task, disregarding the broad practicality of multitasking, or fail to consider the complementary features from audio modality for explicit enhancement. Inspired by the biological information sharing within multiple tasks, in this study, we propose a powerful multi-feature oriented sound source localization and classification framework based on SNNs, namely SpikSLC-Net. Specifically, we design a novel Spiking Hybrid Attention Fusion (SHAF) mechanism that incorporates spiking self-attention modules and spiking cross-attention modules, which can effectively capture temporal dependencies and align relationships among diverse features. Then, considering the vanilla layer normalization (LN) requires dynamic calculation during runtime and involves a significant amount of floating-point operations, we present a unique training-inference-decoupled LN method (DSLN) for SNNs. To further aggregate the multi-scale audio information, two task-specific heads are introduced for the final direction-of-arrival (DoA) estimation and event class prediction. Experimental results demonstrate that the proposed SpikSLC-Net achieves state-of-the-art performance with only 2 time steps on SLoClas dataset.

## 1 Introduction

The sense of hearing has evolved and been maintained so that organisms can utilize the surrounding sounds not only for communication, but also to glean information about the general acoustic environment (Brown & May, 2005). In the auditory field, the ability to localize and recognize sound sources is one of the fundamental and most crucial skills of animals (Grothe et al., 2010). It is beneficial for them to catch prey, evade predators, and find mates in complex environments. As for humans, it also helps individuals to redirect or avoid attention towards certain sound sources during speech communication. Generally speaking, when the sound is transmitted to our ears, it first undergoes a series of physical processes to generate bioelectric activity. Subsequently, nerve impulses travel through the auditory nerve, propagate along the neural pathway, and reach the central auditory cortex (Alain et al., 2001). Combining previous experiences, we can quickly identify the sound we are currently hearing. At the same time, our auditory systems utilize binaural cues to localize the sound, mainly including interaural time difference (ITD) and interaural intensity difference (IID) (Goodman & Brette, 2010). In deep learning, sound event localization and classification (SELC) involves identifying the direction-of-arrival (DoA) and the category of a sound event through a unified framework. SELC has played an essential role in many applications, such as surveillance (Crocco et al., 2016), robot navigation and human-robot interaction (Grumiaux et al., 2022).

Spiking neural networks (SNNs), known as the third generation of neural networks (Maass, 1997), have been a significant family of neuroscience-oriented intelligent models. Compared to artificial neural networks (ANNs), the characteristics of SNNs include having biologically interpretable neural behaviors, utilizing spike signals for encoding and transmitting information, and operating in an

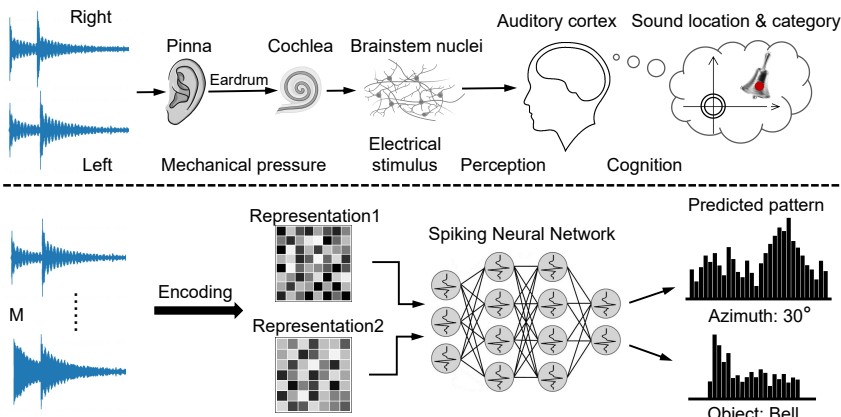

Figure 1: An analogy between the human auditory system and our method. The upper part depicts the pipeline of the human auditory system about sound localization and recognition. The lower part is the corresponding abstract framework based on SNNs, where the color shade in representation and the bar height in predicted pattern signify different spike activity.

event-driven computing mode. Nowadays, SNNs have achieved great success in diverse fields including image classification (Fang et al., 2021; Yao et al., 2021; 2022b; Zhou et al., 2022), object detection (Kim et al., 2020; Su et al., 2023), sound and speech recognition (Zhang & Li, 2021; Yarga et al., 2022; Wang et al., 2023), natural language processing (Lv et al., 2022; Zhu et al., 2023) and so on. However, there are very few works that have demonstrated their effectiveness in SELC task. SNNs inspired by the brain cortex potentially serves as a more energy-efficient and bio-plausible way to perform such multi-task learning. Pan et al. (2021) and Chen et al. (2023) design very complex SNN encoding schemes for sound source localization, but they are difficult to apply in sound recognition. To some extent, they failed to consider the possibility for improvement in the performance of the two tasks through multi-feature fusion and multi-task learning. Besides, the works of SNNs mostly focus on single modality at present (Zhu et al., 2022; Zhou et al., 2022; Cai et al., 2023), lacking the studies on multimodal fusion methods for full SNNs. What's more, the normalization technique shows great advantages in ANNs and its effectiveness in SNNs is also worth exploring. Among these methods, layer normalization (LN) can accelerate training and improve generalization accuracy (Ba et al., 2016). Nevertheless, the normalization approaches proposed for SNNs have largely been developed based on batch normalization (BN) (Wu et al., 2019; Zheng et al., 2021; Kim & Panda, 2021; Duan et al., 2022). This greatly limits SNNs ability to handle tasks beyond computer vision. In (Zhu et al., 2023; Bal & Sengupta, 2023; Lv et al., 2023), they directly use LN from ANNs for language tasks in SNNs. Whereas, the vanilla LN needs to be calculated dynamically during runtime, which involves a substantial number of floating-point multiplication and division operations. Thus the steps do not conform to the calculation characteristics of SNNs.

In this work, we present a unified energy-efficient framework (SpikSLC-Net) composed of SNNs for SELC task with single source. As shown in Fig.1, SpikSLC-Net is motivated by the human auditory system (Yang & Zheng, 2022) and it is able to locate sound position and recognize sound category simultaneously. The framework takes full advantage of multi-feature fusion and multi-task learning, improving the performance of both tasks. We first leverage a learnable spike coding scheme to represent the features of general cross-correlation phase transform (GCC-PHAT) and log-Mel spectrogram from the original audio, respectively. The former is a commonly used feature in sound source localization, while the latter is a commonly used feature in sound recognition. In order to integrate above features, we extend Spiking Self Attention (SSA) (Zhou et al., 2022) to Spiking Cross Attention (SCA) in terms of the properties of SNNs. SCA introduces cross-attention mechanism to SNNs for the first time. In SCA, the Query, Key, and Value are in spike form which only contains 0 and 1, and the softmax operation is discarded. Based on SSA and SCA, we develop the Spiking Hybrid Attention Fusion (SHAF) module to learn intra-modal temporal dependencies and inter-modal alignment. Due to the suitability of LN in processing audio temporal data, we propose a training-inference-decoupled LN method (DSLN) for SNNs, which only calculates the mean while keeping the variance fixed during inference. And we apply a re-parameterization technique to fold the trained

DSLN into the weights so that no additional computation and energy consumption are generated. In brief, our contributions are summarized as follows:

- We present an end-to-end framework based on SNNs (SpikSLC-Net) for joint sound source localization and classification. To the best of our knowledge, this is the first work to implement audio-related multi-task learning by full SNNs.

- We develop a novel spike-form cross-attention named Spiking Cross Attention (SCA) to handle information from different modalities. Based on them, we design a novel Spiking Hybrid Attention Fusion (SHAF) mechanism that can effectively integrate acoustic features including GCC-PHAT and log-Mel spectrogram.

- We propose a training-inference-decoupled LN method (DSLN) for the properties of SNNs. To further eliminate the induced time cost and computational energy consumption of DSLN, we apply a re-parameterization technique to fold the trained DSLN into other layers.

- The experiments conducted on the SLoClas dataset demonstrate that our proposed method outperforms the previous methods. It is worth noting that we achieved more than 97% accuracy in DoA and more than 99% accuracy in SEC tasks with extremely fewer time steps (only 2 time steps).

## 2 RELATED WORK

**Sound source localization and classification.** The general principle of sound source localization (SSL) and sound event classification (SEC) methods is that a multichannel input signal recorded with a microphone array is processed by a feature extraction module and then the features are fed into a neural network, which delivers an estimate of DoA and category (Grumiaux et al., 2022). Adavanne et al. (2018a) proposed a convolutional recurrent neural network to simultaneously recognize and localize sound events based on regression. Shimada et al. (2022) proposed auxiliary duplicating permutation invariant training that enables the model to solve the cases with overlaps. Hu et al. (2023) designed a binaural system with temporal attention for robust localisation of sound sources in noisy conditions. For SNNs, current studies focus on a single task like SSL. Pan et al. (2021) put forward a computational model for SSL under SNN framework, and the core is a Multi-Tone Phase Coding scheme. Chen et al. (2023) introduced a hybrid neural coding framework that incorporated multiple neural coding schemes. And they leveraged a pre-trained ANN to supervise the training of a SNN. This paper concentrates on verifying the effectiveness of multi-task learning in SNNs and developing a powerful spiking framework for SELC task.

**Spiking attention mechanism.** Attention mechanism plays a crucial role in network design as it enhances representation by paying more attention to underlying regions of interest. Compared with ANNs, only a few studies have employed attention modules in SNNs, especially in the multimodal learning field. Yao et al. (2021) integrated a temporal-wise attention module into SNNs to judge the significance of frames and discard the irrelevant frames at the inference stage. Most of them concentrated on attention mechanisms in one or two dimensions (Zhu et al., 2022; Yu et al., 2022; Kundu et al., 2021). Zhou et al. (2022) proposed a novel SSA module in SNNs which is an effective and computation-efficient self-attention variant. Yao et al. (2022a) and Cai et al. (2022) merged multi-dimensional attention with SNNs, including temporal, channel and spatial dimensions. Although these methods are helpful for improving model performance, their application in processing of multimodal features is still unknown. In this paper, we will explore the feasibility of implementing cross-attention and hybrid attention in SNNs for feature fusion.

**Normalization in SNNs.** There are many normalization methods in ANNs, including batch normalization (BN) (Ioffe & Szegedy, 2015), layer normalization (LN) (Ba et al., 2016), group normalization (Wu & He, 2018), instance normalization (Ulyanov et al., 2016), etc. In the SNN field, some works also modify and utilize normalization methods. NeuNorm (Zheng et al., 2021) normalized the feature data along the channel dimension. Threshold-dependent batch normalization (tdBN) (Zheng et al., 2021) extended the scope of BN to the additional temporal dimension. Temporal batch normalization through time (BNTT) (Kim & Panda, 2021) and temporal effective batch normalization (TEBN) (Duan et al., 2022) regulated the data flow utilizing different parameters through time steps. In language tasks, current works just directly adopted vanilla LN used in ANNs to normalize sequence features (Zhu et al., 2023; Bal & Sengupta, 2023; Lv et al., 2023). Nevertheless, their

parameters can not be folded into the weights, so they will increase the computations and running time during inference. Inspired by these methods, we aim to develop a modified LN method with low computations for audio frame processing and make use of the temporal distribution of presynaptic inputs.

## 3 PRELIMINARY

### 3.1 SPIKING NEURON

Different from ANNs, SNNs use sparse binary spike trains to transmit and represent information. In this study, we employ the Leaky Integrate-and-Fire (LIF) neuron, one of the most widely used biological neurons in SNNs. The common form of the LIF neuron is described as:

$$\tau_m \frac{\mathrm{d}u_m(t)}{\mathrm{d}t} = -\left(u_m(t) - u_{\mathrm{rest}}\right) + X_t, \tag{1}$$

where $u_m(t)$ represents the membrane potential of the neuron at time $t$, $X_t$ represents the input from the presynaptic neuron. $\tau_m$ is the membrane time as a constant value that controls the decay, and $u_{\mathrm{rest}}$ is the resting potential after firing. As claimed in previous works (Wu et al., 2018), we convert the above continuous differential equation into a discrete iterative version for numerical simulations:

$$h_i^{t+1,n} = u_i^{t,n} + \frac{1}{\tau_m}\left(-\left(u_i^{t,n} - u_{\mathrm{rest}}\right) + x_i^{t,n}\right), \tag{2}$$

$$o_i^{t+1,n} = \Theta\left(h_i^{t+1,n} - V_{\mathrm{th}}\right), \tag{3}$$

$$u_i^{t+1,n} = o_i^{t+1,n} u_{\mathrm{rest}} + \left(1 - o_i^{t+1,n}\right) h_i^{t+1,n}. \tag{4}$$

Here $h_i^{t,n}$ and $u_i^{t,n}$ denote the value of membrane potential after neural dynamics and after generating a spike at time step $t$ of $i$-$th$ neuron in $n$-$th$ layer, respectively. $o_i^{t,n}$ denotes the spike output at time step $t$. $V_{\mathrm{th}}$ is the voltage threshold and $\Theta(\cdot)$ is the Heaviside step function.

### 3.2 LAYER NORMALIZATION

Layer normalization (LN) is a technique to normalize the distributions of intermediate layers, especially suitable for processing variable-length temporal data. It enables smoother gradients, faster training, and better generalization accuracy, thus having been widely used in ANNs like Transformer (Vaswani et al., 2017). Considering an analog neuron with input $\mathbf{x} = \{\boldsymbol{x}_1, \boldsymbol{x}_2, \ldots, \boldsymbol{x}_N\}$, where $\mathbf{x}$ is the vector representation of size $N$. LN re-centers and re-scales the input as follows:

$$\mathrm{LayerNorm}(\mathbf{x}) = \frac{\mathbf{x} - \mu}{\sqrt{(\sigma)^2 + \epsilon}} \odot \gamma + \beta, \tag{5}$$

$$\mu = \frac{1}{N}\sum_{i=1}^{N} x_i, \quad \sigma = \sqrt{\frac{1}{N}\sum_{i=1}^{N}(x_i - \mu)^2}, \tag{6}$$

where $\odot$ is the element-wise multiplication between two vectors. $\mu_i^{(t)}$ and $\sigma_i^{(t)}$ are the mean and standard deviation of input. Gain $\gamma$ and bias $\beta$ are learnable linear affine factors with the same dimension $N$. In contrast to batch normalization that holds fixed parameters from training and can be folded during inference, LN needs to dynamically compute statistics (i.e., mean and standard deviation) in the inference stage. While computing $\mu$ is straightforward, evaluating $\sigma$ requires the square-root function.

## 4 METHOD

We describe our method for joint sound localization and classification in the following. We first present the audio encoding scheme that convert the acoustic features into spike trains. Next, we depict the spiking hybrid attention fusion block and its vital components like cross-attention mechanism. Then, we introduce the training-inference-decoupled spiking layer normalization method and derive its re-parameterization process. The entire training, re-parameterization, and inference procedure are summarized in Appendix A.

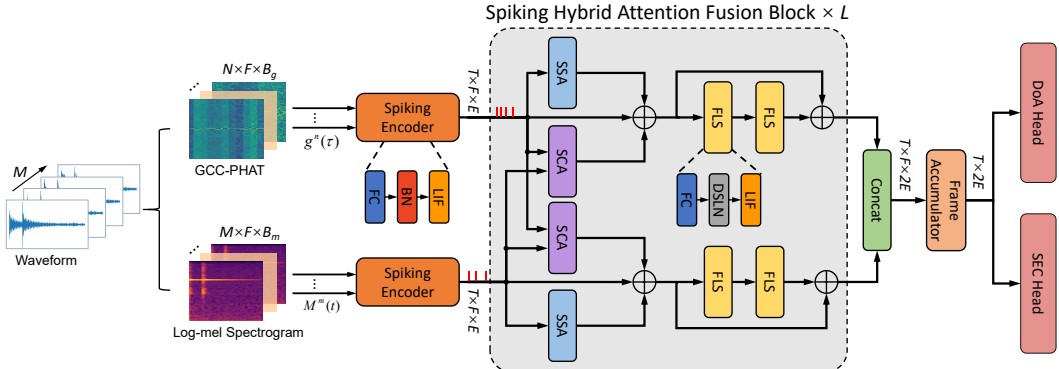

Figure 2: The overview of our proposed SpikSLC-Net for joint sound localization and classification ($\oplus$ indicates the element-wise addition). SpikSLC-Net mainly consists of spiking encoders, hybrid attention fusion blocks, and task-specific heads. Firstly, the GCC-PHAT $\mathbf{g}^n(\tau)$ and log-Mel spectrogram $\mathbf{M}^m(t)$ features from the original multichannel audio are converted into spiking representation by Spiking Encoders. Then, several SHAF Blocks that have SSA, SCA, and DSLN modules are applied to integrate different features. The resulting cross-attentive features are concatenated and accumulated to backend DoA classifier and SEC classifier. $M$, $N$, $F$ and $E$ represent the audio channels number, the total microphone pairs number, the frames number and the embedding dimension respectively. $B_g$ and $B_m$ denote the feature dimension of each frame in corresponding streams.

## 4.1 SPIKING AUDIO ENCODING SCHEME

In sound source localization, time-delay based methods have achieved remarkable success due to their simplicity and effectiveness in computation, especially the GCC-PHAT. It is widely used to estimate the TDoA between any two microphones. Because GCC-PHAT is robust to noise and room reverberations (Florencio et al., 2008), we utilize it as one of the acoustic features. We denote $\mathbf{S}_{p1}$ and $\mathbf{S}_{p2}$ as the STFT of short-time audio signals at a microphone pair $\{(p_1, p_2), \forall p_1 < p_2 \leq M\}$ where $M$ is the total number of microphones. We use $n \leq N$ to index the microphone pair with $N$ the total pair number. Then, the GCC-PHAT feature at time delay $\tau$ is calculated as:

$$\mathbf{g}^n(\tau) = \sum_k \mathcal{R}\left(\frac{\mathbf{S}_{p_1}(k)\mathbf{S}_{p_2}^*(k)}{\left|\mathbf{S}_{p_1}(k)\mathbf{S}_{p_2}^*(k)\right|}e^{j\frac{2\pi k}{N_s}\tau}\right), \tag{7}$$

where $j$ denotes the imaginary unit, $*$ indicates the complex conjugate, $k$ represents the frequency bin, and $N_s$ is the STFT length. The delay lag between two signals arrived, denoted as $\tau$, is reflected in the steering vector $e^{j\frac{2\pi k}{N_s}\tau}$ in Eq.7.

The GCC-PHAT feature is not optimal for SEC because it sums over the entire frequency range without taking into account the sparsity of sound signals in the frequency domain (Qian et al., 2021). In order to explore frequency-domain feature and provide a compact representation, we incorporate log-Mel spectrogram by applying STFT through a mel filter bank. We use $M^m$ to denote the feature from the $m$-th microphone.

Typically, considering the spatio-temporal feature of SNNs, the features are copied and used as the input frame for each time step. Hence, we apply direct encoding with LIF neurons to encode the aforementioned acoustic features into spike trains (Rathi & Roy, 2021). The Spiking Encoder in the first part composed of LIF neurons acts as both the feature extractor and the spike-generator.

## 4.2 SPIKING HYBRID ATTENTION FUSION BLOCK

Here, we design a scalable deep module to refine the multi-feature cues by considering the audio frame variations across different features. Self-attention can exploit the overall situation instead of a single frame, which is particularly useful for sequential signal processing like audio. Cross-attention can utilize the synchronization information between different feature modalities. Therefore, we integrate these two attention mechanisms related to SNNs into our SHAF block to achieve maximum utilization of input features.

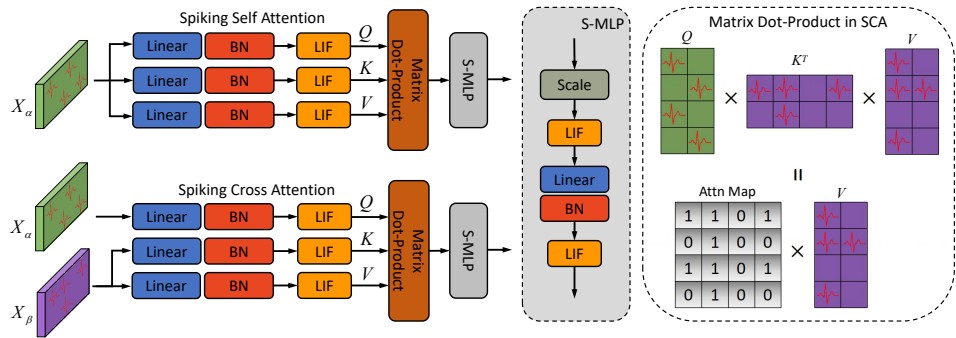

Figure 3: Illustration of Spiking Self Attention (SSA) and Spiking Cross Attention (SCA) mechanisms. A red spike denotes a value of 1 at that location. SSA is appropriate for handling single features, while SCA is suitable for processing multiple features. And SCA inherits the advantages of SSA's low energy consumption and decomposability.

We adopt Spiking Self Attention (SSA) (Zhou et al., 2022) as a fundamental module and extend SSA to Spiking Cross Attention (SCA). For SCA, as illustrated in Fig.3 left part, given two spike input feature sequences $X_\alpha \in \mathbb{R}^{T \times F \times E}$ and $X_\beta \in \mathbb{R}^{T \times F \times E}$, the float-point query ($Q$), key ($K$), and value ($V$) are computed by three learnable matrices firstly. $Q$ corresponds to $X_\alpha$, while $K$ and $V$ correspond to $X_\beta$. Note, the linear operation here is only addition, because the input is a spike tensor. Then they are converted into spiking sequences via spike neuron layers $\mathcal{SN}(\cdot)$:

$$Q_\alpha = \mathcal{SN}_Q\left(\mathrm{BN}\left(X_\alpha W_Q\right)\right), K_\beta = \mathcal{SN}_K\left(\mathrm{BN}\left(X_\beta W_K\right)\right), V_\beta = \mathcal{SN}_V\left(\mathrm{BN}\left(X_\beta W_V\right)\right) \quad (8)$$

where $Q, K, V \in \mathbb{R}^{T \times F \times E}$, $\mathrm{BN}(\cdot)$ represents the batch normalization operation. Then we add a scaling factor $s$ to control the large value of the matrix multiplication result. The spike-friendly cross-attention we use is defined as:

$$\mathrm{SCA}'(X_\alpha, X_\beta) = \mathcal{SN}\left(Q_\alpha K_\beta^{\mathrm{T}} V_\beta * s\right), \quad (9)$$

$$\mathrm{SCA}(X_\alpha, X_\beta) = \mathcal{SN}\left(\mathrm{BN}\left(\mathrm{Linear}\left(\mathrm{SCA}'(X_\alpha, X_\beta)\right)\right)\right). \quad (10)$$

SCA is independently conducted on each time step and can easily be extended to the multi-head SCA.

As shown in Fig.2, we jointly model the co-occurrence and synchrony of the different features at frame level in the SHAF block through the SSA and SCA modules. The overall SHAF block models the interactions in a symmetrical manner. In the following, to simplify the explanation, we only introduce half of the block which includes one SSA, one SCA and two FLS modules. The intermediate SHAF features are formulated as:

$$\tilde{\mathbf{X}}_{\alpha,\beta} = \mathbf{X}_\alpha + \mathrm{SSA}\left(\mathbf{X}_\alpha\right) + \mathrm{SCA}\left(\mathbf{X}_\alpha, \mathbf{X}_\beta\right) \quad (11)$$

where the skip connections are essential for preserving the identity information from the input stream $\mathbf{X}_\alpha$. The final SHAF output is computed as:

$$\mathrm{FLS}\left(\tilde{\mathbf{X}}_{\alpha,\beta}\right) = \mathcal{SN}\left(\mathrm{FC}\left(\mathrm{DSLN}\left(\tilde{\mathbf{X}}_{\alpha,\beta}\right)\right)\right) \quad (12)$$

$$\mathrm{SHAF}\left(\mathbf{X}_\alpha, \mathbf{X}_\beta\right) = \tilde{\mathbf{X}}_{\alpha,\beta} + \mathrm{FLS}\left(\mathrm{FLS}\left(\tilde{\mathbf{X}}_{\alpha,\beta}\right)\right) \quad (13)$$

where DSLN denotes our proposed training-inference-decoupled spiking LN method in Section 4.3. With two input feature streams, SHAF consists of two above parallel units in practice.

## 4.3 DECOUPLED SPIKING LAYER NORMALIZATION

As we known, vanilla layer normalization (LN) needs to calculate the mean and standard deviation in real time during inference, thus it can't be folded into the parameters of previous layers. This problem causes LN to be very unfriendly to SNNs, because it brings a large number of floating-point multiplication and division operations, and increasing the energy consumption of SNNs. Moreover, most neuromorphic chips do not support this kind of operation (Furber et al., 2014; Akopyan et al.,

2015; Davies et al., 2018), making it difficult to deploy such SNNs on them. To this end, we propose Decoupled Spiking Layer Normalization (DSLN), which uses a fixed variance and dynamically calculate the mean in the stage of inference. Given that $\boldsymbol{x}_i^{l-1}[t] \in \mathbb{R}^N$ is the $i$-th vector of the input sequence to layer $l$ at time step $t$, the spiking neuron with DSLN can be described as:

$$
\begin{aligned}
\boldsymbol{x}_i^{l-1}[t] &= \boldsymbol{W}^l \boldsymbol{o}_i^{l-1}[t] + b, \\
\boldsymbol{u}_i^l[t] &= \tau \boldsymbol{u}_i^l[t-1]\left(1 - \boldsymbol{o}_i^l[t-1]\right) + \hat{\boldsymbol{x}}_i^{l-1}[t], \\
\text{where} \quad \hat{\boldsymbol{x}}_i[t] &= \text{DSLN}(\boldsymbol{x}_i[t]) = \frac{\boldsymbol{x}_i[t] - \boldsymbol{\mu}_i[t]}{\sqrt{\boldsymbol{\sigma}_i^2 + \epsilon}} \odot \boldsymbol{\gamma} + \boldsymbol{\beta}.
\end{aligned}
\tag{14}
$$

Here, $\odot$ is the element-wise multiplication between two vectors. We omit $u_{\text{rest}}$ and the input decay to simplify the equation. $\boldsymbol{W}^l$ is the synaptic weights between layer $l-1$ and layer $l$, and $b$ is the bias. $\boldsymbol{u}_i^l[t]$ and $\boldsymbol{o}_i^l[t]$ are the membrane potential and output spikes of $i$-th corresponding neurons in layer $l$ at time step $t$. Gain $\gamma$ and bias $\beta$ are learnable factors with the same dimension $N$. We calculate the mean $\boldsymbol{\mu}$ at each time step, while calculate the variance $\boldsymbol{\sigma}^2$ from samples at all time steps. During inference, we apply the expectation of the variance and fix it, but the mean needs to be calculated dynamically.

In order to eliminate the extra time cost and computational energy consumption caused by DSLN, we fold it into the previous linear layer. The above equation is reorganized as:

$$
\begin{aligned}
\text{DSLN}(\boldsymbol{x}_i[t]) &= \frac{\boldsymbol{\gamma}}{\sqrt{\boldsymbol{\sigma}_i^2 + \epsilon}} \odot \boldsymbol{x}_i[t] + \boldsymbol{\beta} - \frac{\boldsymbol{\gamma}}{\sqrt{\boldsymbol{\sigma}_i^2 + \epsilon}} \boldsymbol{\mu}_i[t], & \boldsymbol{x}_i[t] \in \mathbb{R}^m \\
&= A_i \odot \boldsymbol{x}_i[t] + \boldsymbol{\beta} - A_i \frac{1}{n} \sum_{j=1}^{n} \boldsymbol{x}_{i,j}[t], & A_i \in \mathbb{R}^n \\
&= A_i \odot \boldsymbol{W} \boldsymbol{o}_i[t] + A_i \odot b + \boldsymbol{\beta} - \boldsymbol{W}'_{1,i} \boldsymbol{x}_i[t], & \boldsymbol{W} \in \mathbb{R}^{n \times m}, \boldsymbol{W}'_{1,i} \in \mathbb{R}^{n \times n} \\
&= (\boldsymbol{W}'_{2,i} - \boldsymbol{W}'_{1,i} \boldsymbol{W}) \boldsymbol{o}_i[t] + A_i \odot b + \boldsymbol{\beta} - \boldsymbol{W}'_{1,i} b, & \boldsymbol{W}'_{2,i} \in \mathbb{R}^{n \times m}
\end{aligned}
\tag{15}
$$

where $W$ represents the parameters of the previous linear layer. $m$ and $n$ denote the input channel dimension and output channel dimension of the linear layer, respectively. $A_i$ and $W'_i$ are the expressions of constants that result from previous calculations.

## 4.4 LOSS FUNCTION

As DoAs are spatially continuous, we use a Gaussian-like vector $p(\theta)$ (Adavanne et al., 2018b) to represent the posterior probability likelihoods of a sound presence in the azimuth direction of $\theta$:

$$
p(\theta) = \exp\left(-\frac{|\theta - \tilde{\theta}|^2}{\sigma^2}\right),
\tag{16}
$$

where $\theta \in [1, 360°]$, $\tilde{\theta}$ is the ground truth, and $\sigma$ is a constant related to the width of the Gaussian function. The decoded location of the sound source is the maximum of the classifier output. The MSE loss is used to measure the similarity between $p(\theta)$ and predicted $\hat{p}(\theta)$, computed as:

$$
\mathcal{L}_{SSL} = \|p(\theta) - \hat{p}(\theta)\|_2^2.
\tag{17}
$$

For the SEC task, we adopt one-hot encoding and cross entropy loss. The loss function is computed between the SEC estimate and the ground truth sound event class, formulated as:

$$
\mathcal{L}_{SEC} = -\log\left(\frac{\exp(y(\tilde{c}))}{\sum_{c=1}^{C} \exp(\hat{y}(c))}\right),
\tag{18}
$$

where $\tilde{c}$ is the ground truth label and $c = 1, ..., C$ is the event class label with $C$ as the total class number. $\hat{y}(c)$ is the output of SEC head.

To optimize the final multi-task network, we use a coefficient $\lambda$ to balance the performance of different tasks and the objective function for SELC is as follows:

$$
\mathcal{L}_{multi} = \lambda \mathcal{L}_{SSL} + (1 - \lambda) \mathcal{L}_{SEC}.
\tag{19}
$$

Table 1: Performance comparison between our method and previous works on SLoClas dataset.

| Method | Type | Time Step | DoA | | SEC Acc. (%) |
|---|---|---|---|---|---|
| | | | MAE(°) | Acc.(%) | |
| GCC-PHAT-CNN (Qian et al., 2021) | ANN training | 1 | 4.39 | 86.94 | 80.01 |
| MTPC-CSNN (Pan et al., 2021) | SNN training | 2 | 1.34 | 93.16 | - |
| MTPC-CSNN (Pan et al., 2021) | SNN training | 4 | 1.23 | 93.95 | - |
| MTPC-CSNN (Pan et al., 2021) | SNN training | 8 | 1.02 | 94.72 | - |
| MTPC-RSNN (Pan et al., 2021) | SNN training | 51 | 1.48 | 94.30 | - |
| Hybrid Coding (Chen et al., 2023) | Hybrid training | 4.37 | 0.60 | 95.61 | - |
| SpikSLC-Net-1-128 (Ours) | SNN training | 2 | **0.51** | **97.53** | **99.30** |
| SpikSLC-Net-2-64 (Ours) | SNN training | 2 | **0.55** | **97.26** | **99.30** |
| SpikSLC-Net-2-128 (Ours) | SNN training | 2 | **0.47** | **97.46** | **99.33** |
| SpikSLC-Net-3-128 (Ours) | SNN training | 2 | **0.45** | **97.39** | **99.42** |

## 5 EXPERIMENTS

In this section, we validate the effectiveness of our proposed SpikSLC-Net for SELC task on the dataset used in SNNs. We first compare our algorithm with other state-of-the-art methods to demonstrate the advantages of SpikSLC-Net. After that, we conduct adequate ablation studies to provide a deep understanding of our framework. Finally, we carry out a more comprehensive analysis of the robustness to noisy environment.

### 5.1 EXPERIMENTAL SETUP

**Dataset.** As previous SNN works (Pan et al., 2021; Chen et al., 2023), we conduct the experiments on SLoClas dataset (Qian et al., 2021) which is designed for real-world sound localization and classification tasks. The dataset contains 23.27 hours of data recorded by a 4-channel microphone array. There are a total of 10 types of sounds, each played by a loudspeaker located 1.5 meters away from the microphone array. The azimuth angle ranges from $1°$ to $360°$, with an interval of $5°$. Besides, 6 different types of outdoor noise were recorded at 4 different azimuth angles.

**Evaluation metrics.** We use the mean absolute error (MAE) and the accuracy (Acc) metrics to evaluate the performance of models. For DoA estimation, MAE is calculated as the average angular difference between the ground truth and the predicted DoA of the sound. Acc represents the proportion of correctly predicted samples. They are computed by:

$$\text{MAE}(°) = \frac{1}{N}\sum_{i=1}^{N}\left|\hat{\theta}_i - \tilde{\theta}_i\right|, \quad \text{Acc}_\theta(\%) = \frac{1}{N}\sum_{i=1}^{N}\left(\left|\hat{\theta}_i - \tilde{\theta}_i\right| < \eta\right), \tag{20}$$

where $\hat{\theta}_i$ and $\tilde{\theta}_i$ denote the estimated and the ground truth azimuth angle of sample $i$, respectively. $\eta$ is the DoA error allowance in determining whether a sample has been correctly classified and we set $\eta$ to $2.5°$ in this paper. $N$ is the total number of test samples. For SEC, we use Acc metric and compute as:

$$\text{Acc}_e(\%) = \frac{1}{N}\sum_{i=1}^{N}(\hat{c}_i = \tilde{c}_i), \tag{21}$$

where $\hat{c}_i$ and $\tilde{c}_i$ denote the predicted and the ground truth sound event label sample $i$.

**Implementation details.** All experiments below are carried out on NVIDIA 2080Ti GPUs. The models are implemented based on PyTorch Paszke et al. (2019) and SpikingJelly Fang et al. (2020) frameworks. Our model is optimized by AdamW optimizer Loshchilov & Hutter (2017) with standard settings and the learning rate is $1e^{-4}$. We set the reset value $u_{\text{rest}}$ of LIF neurons to 0, the membrane time constant $\tau$ to 2.0, and the threshold $V_{\text{th}}$ to 1.0. The sampling rate of the audio is set to $16\,\text{kHz}$ and the standard deviation $\sigma$ of the DoA-based posterior probability in Eq.16 is set to 5. We compute the 51-dimensional GCC-PHAT at each $64\,\text{ms}$ segment with the delay lag $\tau \in [-25, 25]$ for each microphone pair. The log-Mel spectrogram is computed with 51 mel-scale filters at a frequency range from $250\,\text{Hz}$ to $8\,\text{kHz}$ and the window size is $64\,\text{ms}$.

## 5.2 COMPARISON TO PRIOR WORK

We compare the performance of the proposed SpikSLC-Net with relevant sound source localization and classification methods in SNNs. Quantitative comparison results of the previous methods and our model on the test set are shown in Tab.1. We try a variety of models with different embedding dimensions and numbers of SHAF blocks. It can be seen that that our proposed method significantly outperform others with only 2 time steps and even exceeds the performance of ANN model. And the performance is improved as the dimensions or blocks increase. The accuracy on the SEC task has approached 100%, which indicates that our model has strong performance on sound classification. It is worth noting that previous works in SNNs only focused on one task, while we can simultaneously perform two tasks.

Table 2: Ablation study results on time step.

| Architecture | Time Step | DoA | | SEC Acc.(%) |
|---|---|---|---|---|
| | | MAE(°) | Acc.(%) | |
| SpikSLC-Net-1-128 | 2 | 0.51 | 97.53 | 99.30 |
| | 4 | 0.51 | 97.51 | 99.40 |
| | 6 | 0.56 | 97.33 | 99.39 |
| SpikSLC-Net-2-128 | 1 | 0.62 | 97.04 | 99.22 |
| | 2 | 0.47 | 97.46 | 99.33 |
| | 4 | 0.46 | 97.46 | 99.35 |
| | 6 | 0.45 | 97.69 | 99.38 |
| SpikSLC-Net-2-64 | 2 | 0.55 | 97.26 | 99.30 |
| SpikSLC-Net-2-256 | 2 | 0.53 | 97.04 | 99.52 |
| SpikSLC-Net-2-512 | 2 | 0.62 | 96.92 | 99.49 |
| SpikSLC-Net-3-128 | 2 | 0.45 | 97.39 | 99.42 |
| SpikSLC-Net-4-64 | 2 | 0.56 | 97.15 | 99.30 |
| SpikSLC-Net-4-128 | 2 | 0.76 | 96.46 | 99.47 |

## 5.3 ABLATION STUDY

**Time steps, numbers of SHAF blocks and dimension of embeddings.** The performance regarding different simulation time steps, numbers of SHAF blocks and dimension of embeddings is shown in Tab.2. It can be found that in the beginning, the MAE of SELC is lower as the value of hyperparameters increase. However, if the number of model parameters is too large, it will occur overfitting on the dataset. The accuracy of SEC does not change much, hyperparameter, which shows that our model has strong ability to extract features that are beneficial to sound classification.

Table 3: Ablation study results on $\lambda$.

| $\lambda$ | DoA | | SEC Acc.(%) |
|---|---|---|---|
| | MAE(°) | Acc.(%) | |
| 0 | - | - | 99.31 |
| 0.800 | 0.56 | 96.93 | 99.34 |
| **0.990** | **0.47** | **97.46** | 99.33 |
| 0.999 | 0.53 | 97.29 | 99.33 |
| 1 | 0.52 | 97.30 | - |

**Effect of loss weight.** In order to verify the effectiveness of multi-task learning in improving the performance of both subtasks, we conducted experiments with different values of $\lambda$. As shown in Tab.3, after adopting multi-task learning, the performance of sound source localization is most obviously improved.

Table 4: Ablation study results on norm methods.

| Norm Method | DoA | | SEC Acc.(%) |
|---|---|---|---|
| | MAE(°) | Acc.(%) | |
| Vanilla LN | 0.47 | 97.41 | 99.30 |
| Variant LN | 0.50 | 97.38 | 99.34 |
| BN1d | 0.50 | 97.40 | 99.30 |
| DSLN | **0.47** | **97.46** | 99.33 |

**Different normalization methods.** As shown in Tab.4, we try different normalization methods in our model. For variant LN, we move the variance from the denominator to the numerator in vanilla LN. It is observed that our DSLN achieves the best performance and does not generate extra computations, compared with other methods.

## 6 CONCLUSION

In this work, we present a framework (SpikSLC-Net) for joint sound source localization and classification based on full SNNs. It successfully implement multi-task learning in SNNs. We design a novel Spiking Hybrid Attention Fusion block that consists of Spiking Cross Attention and Spiking Self Attention modules. The block an effectively integrate different acoustic features. Moreover, we propose a training-inference-decoupled layer normalization method (DSLN) for SNNs. By combining the re-parameterization technique, DSLN does not generate additional computational costs. The evaluation on the test set verifies the effectiveness and efficiency of the our method. We hope our investigations pave the way for further research on audio-related SNN models.

## 7 REPRODUCIBILITY STATEMENT

We have made great efforts to ensure the reproducibility of the results reported in this paper. The experiment settings, evaluation metrics, datasets, and implementation details are clearly presented in Section 5.1.

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

APPENDIX

## A    TRAINING AND INFERENCE PROCEDURE

---

**Algorithm 1** Training, re-parameterization and inference of our SpikSLC-Net.

---

**Training**

**Input**: A SNN with DSLN; training dataset; total training iteration $I_{\text{train}}$.

**Output**: The well-trained SNN.

1: **for** all $i = 1, 2, \ldots, I_{\text{train}}$ **do**
2:     Get mini-batch data $\boldsymbol{x}(i)$, DoA label $\boldsymbol{\theta}(i)$ and event class label $\boldsymbol{c}(i)$;
3:     Feed the $\boldsymbol{x}(i)$ into the original SNN;
4:     Calculate the SNN output $\boldsymbol{o}_\theta(i)$ and $\boldsymbol{o}_c(i)$;
5:     Compute DoA estimation loss $\mathcal{L}_{\text{SSL}} = \mathcal{L}_{\text{MSE}}(\boldsymbol{o}_\theta(i), \boldsymbol{\theta}(i))$ and event classification loss $\mathcal{L}_{\text{SEC}} = \mathcal{L}_{\text{CE}}(\boldsymbol{o}_c(i), \boldsymbol{c}(i))$;
6:     Calculate the derivative of the loss with respect to the weights;
7:     Update the SNN weights ($\mathbf{W} = \mathbf{W} - \eta \frac{\partial L}{\partial \mathbf{W}}$) where $\eta$ is learning rate. (see Appendix B)
8: **end for**

**Re-parameterization**

**Input**: The trained SNN with DSLN; total number of DSLN $n$.

**Output**: The re-parameterized SNN without DSLN.

1: **for** all $i = 1, 2, \ldots, n$ **do**
2:     Fold the parameters of $i$-th DSLN into $i$-th $W_{\text{th}}$;
3: **end for**

**Inference**

**Input**: The re-parameterized trained SNN; test dataset; total test iteration $I_{\text{test}}$.

**Output**: The output.

1: **for** all $i = 1, 2, \ldots, I_{\text{test}}$ **do**
2:     Feed the $\boldsymbol{x}(i)$ into the re-parameterized SNN;
3:     Calculate the SNN output $\boldsymbol{o}_\theta(i)$ and $\boldsymbol{o}_c(i)$;
4:     Compare the DoA estimation result $\boldsymbol{o}_\theta(i)$ and $\boldsymbol{\theta}(i)$ .
5:     Compare the event classification result $\boldsymbol{o}_c(i)$ and $\boldsymbol{c}(i)$.
6: **end for**

---

## B    TRAINING STRATEGY FOR SNNS

The integration and firing behavior of SNN neurons will result in the non-differentiability of the transfer function. So it is difficult to apply standard backpropagation in the training phase. Recently, methods based on surrogate gradient have provided an effective solution for training deep SNNs. Here, we choose Spatio-Temporal Backpropagation (STBP) (Wu et al., 2018) as our training method and adopt the rectangular function to approximate the derivative of spike activity. With $L$ representing the loss function, the gradients $\partial L/\partial o_i^{t,n}$ and $\partial L/\partial u_i^{t,n}$ can be computed as follows:

$$
\begin{cases}
\dfrac{\partial L}{\partial o_i^{t,n}} = \displaystyle\sum_j \dfrac{\partial L}{\partial u_j^{t,n+1}} \dfrac{\partial u_j^{t,n+1}}{\partial o_i^{t,n}} + \dfrac{\partial L}{\partial u_i^{t+1,n}} \dfrac{\partial u_i^{t+1,n}}{\partial o_i^{t,n}} \\[3mm]
\dfrac{\partial L}{\partial u_i^{t,n}} = \dfrac{\partial L}{\partial o_i^{t,n}} \dfrac{\partial o_i^{t,n}}{\partial u_i^{t,n}} + \dfrac{\partial L}{\partial u_i^{t+1,n}} \dfrac{\partial u_i^{t+1,n}}{\partial u_i^{t,n}} \\[3mm]
\nabla w_{ji}^n = \displaystyle\sum_{t=1}^{T} \dfrac{\partial L}{\partial u_j^{t,n+1}} o_i^{t,n}
\end{cases}
\tag{22}
$$

Due to the non-differentiable property of the binary spike activities, $\partial o_k/\partial u_k$ cannot be derived. We adopt shifted ArcTan function $h(u)$ to approximate the derivative of spike activity, which is defined

by

$$h(u) = \frac{1}{\pi} \arctan(\pi u) + \frac{1}{2} \tag{23}$$

## C  ADDITIONAL RESULTS

Since there are limited datasets and models for joint sound localization and classification, we construct a suitable dataset named DCASE 2019 Part for further experiments. The dataset is the part of the DCASE 2019 dataset without overlapping sound sources. It has 11 sound sources and the azimuth angle interval is 10 degrees. For traditional ANN methods, we employ a modified version of EINV2 (Cao et al., 2021) which is originally used in sound event localization and detection and implement an ANN version of our model (SLC-Net). When evaluating models on DCASE 2019 Part dataset, the threshold for accuracy calculation is set at 5 degrees, and it is set to 2.5 degrees on SLoClas dataset. The results are shown in Tab. 5. As we can see, our method outperforms the ANN methods on both sound event localization and classification tasks.

Table 5: Performance comparison between our method and previous ANN works on different dataset.

| Method | Dataset | Type | DoA | | SEC Acc. (%) |
|---|---|---|---|---|---|
| | | | MAE(°) | Acc.(%) | |
| EINV2 (Cao et al., 2021) | SLoClas | ANN | 0.80 | 96.92 | 98.08 |
| SLC-Net-2-128 | SLoClas | ANN | 0.87 | 96.52 | 99.11 |
| SpikSLC-Net-1-128 (Ours) | SLoClas | SNN | **0.45** | **97.39** | **99.42** |
| EINV2 (Cao et al., 2021) | DCASE2019 Part | ANN | 7.27 | 55.29 | 99.77 |
| SLC-Net-2-128 | DCASE2019 Part | ANN | 3.89 | 80.42 | 99.88 |
| SpikSLC-Net-1-64 (Ours) | DCASE2019 Part | SNN | **2.42** | **86.08** | **1.00** |

