# OpenReview forum: "Spiking Hybrid Attentive Mechanism with Decoupled Layer Normalization for Joint Sound Localization and Classification"
_ICLR.cc/2024/Conference — Submitted to ICLR 2024_

### Official Review · Reviewer_teCi · 2023-10-29

**Soundness:** 3 good
**Presentation:** 3 good
**Contribution:** 2 fair
**Rating:** 5
**Confidence:** 4

**Summary:**

The authors propose SpikSLC-Net, a novel SNN based architecture dealing with sound event localization and classification tasks simultaneously. To effectively integrate acoustic features extracted jointly from both GCC-PHAT and Log-mel Spectrogram,  they combine SSA module together with its extended SCA module and propose a novel SHAF block involving the aforementioned two attention modules, which is declared as fundamental to utilize synchronization information between multiple feature modalities. Experiments also manifest a state-of-the-art performance of the proposed architecture.

**Strengths:**

1. To the best of my knowledge, it is the first work for a SNN based model manifesting its strong effectiveness in multitasking like SELC task, even exceeds the performance of ANN model.
2. The paper is well written and clearly presented. The motivation is meaningful and interesting.
3. Employing GCC-PHAT feature and Log-mel Spectrogram altogether to fully explore the characteristic of sound sources is advisable, which naturally fit for the design of multi-head attention mechanism.

**Weaknesses:**

1. Though achieving state-of-the-art performance on SLoClas dataset, the proposed architecture still lacks novelty. According to my understanding, the core part of the architecture is the SHAF block, however its components are just copies or slight modifications/extensions of spiking self attention mechanism proposed in [zhou et al., 2022]. The extension seems straight forward and may not be regarded as a genuine technical contribution.
2. Some of the notations are not align with the others. For example, sometimes the embedding feature dimension is denoted as $D$, but sometimes not (fig 2).
3. Though performing best on SLoClas dataset, it is hard to conclude that the current work is better than previous works in terms of sound localization and classification. More experiments on various datasets is recommended.
4. Comprehensive analysis of the robustness to noisy environment is insufficient.
5. As in [zhou et al., 2022], comparison of computation complexity and estimated power consumption are highly recommended to be embraced in the experiments.

**Questions:**

The above

---

> ### Author Response · Authors · 2023-11-22
> **Response to Reviewer teCi (Part 1/2)**
>
> Thanks for your constructive and valuable feedback. We would like to address your concerns below.
>
> >Though achieving state-of-the-art performance on SLoClas dataset, the proposed architecture still lacks novelty. According to my understanding, the core part of the architecture is the SHAF block, however its components are just copies or slight modifications/extensions of spiking self attention mechanism proposed in [1]. The extension seems straight forward and may not be regarded as a genuine technical contribution.
>
> **R:** While the SCA module in SHAF block is inspired by SSA module proposed in [1], their input types and functionalities are completely different. The SSA, based on the spiking self-attention mechanism, is utilized to model long-range interdependencies within the same input. But the SCA applies the spiking cross-attention mechanism to align features between different input stream. To the best of our knowledge, this is the first work to explore the effectiveness of cross-attention mechanism in SNNs. We believe that this investigation not only promotes the application of SNN in the audio field, but also paves the way for further research on multimodal SNN models.
>
> Moreover, the whole SHAF block employs a hybrid attention mechanism, instead of a single self-attention mechanism or a single cross-attention mechanism. The block has strong ability in learning intra-modal acoustic temporal dependencies and aligning inter-modal acoustic relationships. Each component of the block has a significant impact on the model performance.
>
> In addition, the DSLN method we proposed is also a core part of the architecture. Different from vanilla LayerNorm, DSLN only calculates the mean while keeping the variance fixed during inference. And we apply a re-parameterization technique to fold the trained DSLN into the weights so that no additional computation and energy consumption are generated.
>
> >Some of the notations are not align with the others. For example, sometimes the embedding feature dimension is denoted as, but sometimes not (Fig. 2).
>
> **R:** We will correct them in the revised version, thank you for your help!
>
> >Though performing best on SLoClas dataset, it is hard to conclude that the current work is better than previous works in terms of sound localization and classification. More experiments on various datasets is recommended.
>
> **R:** Since there are limited datasets and models for joint sound localization and classification, we construct a suitable dataset named DCASE 2019 Part. The dataset is the part of the DCASE 2019 dataset without overlapping sound sources. It has 11 sound sources and the azimuth angle interval is 10 degrees.
> Considering previous SNN methods have not made their code available, we resort to using traditional ANN methods for comparison purposes. Specifically, we utilize a modified version of EINV2 [2] which is originally designed for sound event detection and implement an ANN version of our model (SLC-Net).
> When evaluating models on DCASE 2019 Part dataset, the threshold for accuracy calculation is set at 5 degrees, and it is set to 2.5 degrees on SLoClas dataset.
> The results are shown in Tab. R4-1. As we can see, our method outperforms other methods on both sound event localization and classification tasks.
>
> | Method | Dataset | DoA MAE(&deg;) | DoA Acc(\%) | SEC Acc(\%) |
> | :---- | :---- | :----: | :----: | :----: |
> | GCC-PHAT-CNN | SLoClas |4.88 | 84.50 | 78.90 |
> | EINV2 | SLoClas |0.80 | 96.92 | 98.08 |
> | SLC-Net-2-128 | SLoClas | 0.87 | 96.52 | 99.11 |
> | SpikSLC-Net-3-128 (Ours) | SLoClas | **0.45** | **97.39** | **99.42** |
> | SLC-Net-2-128 | DCASE2019 Part | 3.89 | 80.42 | 99.88 |
> | EINV2 | DCASE2019 Part |7.27 | 55.29 | 99.77 |
> | SpikSLC-Net-1-64 (Ours) | DCASE2019 Part | **2.42**| **86.08** | **1.00** |
> **Table R4-1: Comparison of different methods.**

---

> ### Author Response · Authors · 2023-11-22
> **Response to Reviewer teCi (Part 2/2)**
>
> >Comprehensive analysis of the robustness to noisy environment is insufficient.
>
> **R:** Here, we evaluate the noise robustness of our model under various noisy conditions. We select the well-trained SpikSLC-Net-2-128 on SLoClas dataset as the testing model. The noise sounds are from SLoClas dataset, which contains the recordings of 6 types of noise at 4 DoAs. We randomly select the noise audio from one of the four directions, 0&deg;, 90&deg;, 180&deg;, 270&deg;.
> We regard the noise as the addictive background noise. In order to simulate the background noise from all directions, we add the noise sound clip to microphone signal in the channel dimension.
> In Tab. R4-2, we show the results on noisy data of various Signal-to-Noise ratio (SNR). It is understandable that MAE decreases as SNR increases across all conditions. For high SNR test data, our model performs reasonably well.
>
> | SNR (dB)| DoA MAE(&deg;) | DoA Acc(\%) | SEC Acc(\%) |
> | :---- | :----: | :----: | :----: |
> | -10| 48.62 | 35.27 | 56.46 |
> | 0 | 31.75| 56.73 | 71.77 |
> | 5 | 24.20 | 66.62 | 79.57 |
> | 10 | 16.53| 76.25 | 87.37 |
> | 15 | 9.74 | 85.17 | 93.43 |
> | 20 | 4.59 |92.17 | 97.25 |
> **Table R4-2: Analysis of the robustness to noisy environment.**
>
> >As in [1], comparison of computation complexity and estimated power consumption are highly recommended to be embraced in the experiments.
>
> **R:** The contributions of previous SNN methods focus on coding methods rather than models, so it is difficult to make a fair comparison of algorithm complexity. Furthermore, other methods cannot perform both sound source localization and recognition simultaneously. Thus, we only provide the computation complexity and estimated power consumption of our models and other ANN models.
> In SNNs, the synaptic operations (SOPs) are used to describe the computation complexity, which is calculated by
>
> $\operatorname{SOPs}(l)=fr \times T \times \operatorname{FLOPs}(l)$,
>
> where $l$ is a layer in SpikSLC-Net and $fr$ represents the firing rate of the input spike train, and $T$ denotes the time step. $\operatorname{FLOPs}(l)$ refers to floating point operations of $l$, which is the number of multiply-and-accumulate (MAC) operations. And SOPs is the number of spike-based accumulate (AC) operations.
> Refer to previous works [1][3], we assume that the operations are implemented on the 45nm hardware [4], where the theoretical energy consumption $EMAC = 4.6pJ$ and $EAC = 0.9pJ$.
>
> The results are shown in Tab. R4-3. From the results, it can be seen that the OPs and the theoretical energy consumption of our models are much lower compared with ANN models that have good performance.
>
>
> | Method | Param (M)|OPs (G) | Power (mJ) |
> | :---- | :----: | :----: | :----: |
> | GCC-PHAT-CNN | 4.17 | 0.048 | 0.143 |
> |EINV2 | 85.45 | 8.96 | 20.562 |
> |SLC-Net-2-128 | 4.15 | 0.067 | 0.152 |
> |SpikSLC-Net-1-128 (Ours) | **3.61** | **0.017** | **0.031** |
> | SpikSLC-Net-3-128 (Ours) | 4.68 | 0.056 | 0.066 |
> **Table R4-3: Comparison of computation complexity and estimated power consumption.**
>
>
> We will add these promising comparison results to the revised version.
>
>
>
> Reference:
>
> [1] Zhaokun Zhou, Yuesheng Zhu, Chao He, Yaowei Wang, Shuicheng Yan, Yonghong Tian, and Li Yuan. Spikformer: When spiking neural network meets transformer. arXiv preprint arXiv:2209.15425, 2022.
>
> [2] Cao, Yin and Iqbal, Turab and Kong, Qiuqiang and An, Fengyan and Wang, Wenwu and Plumbley, Mark D. An improved event-independent network for polyphonic sound event localization and detection. ICASSP 2021-2021 IEEE International Conference on Acoustics, Speech and Signal Processing (ICASSP).
>
> [3] Man Yao, Guangshe Zhao, Hengyu Zhang, Yifan Hu, Lei Deng, Yonghong Tian, Bo Xu, and Guoqi Li. Attention spiking neural networks. arXiv preprint arXiv:2209.13929, 2022.
>
> [4] Mark Horowitz. 1.1 computing’s energy problem (and what we can do about it). In 2014 IEEE International Solid-State Circuits Conference Digest of Technical Papers (ISSCC), pages 10–14. IEEE, 2014.

---

### Official Review · Reviewer_zvoL · 2023-10-30

**Soundness:** 1 poor
**Presentation:** 1 poor
**Contribution:** 1 poor
**Rating:** 1
**Confidence:** 3

**Summary:**

The paper presents a spiking neural network model (SpikSLC-Net) for joint sound localization and classification. Extending earlier work on attention in spiking neural networks, the authors propose a cross-attention mechanism. In addition, the propose a form of layer normalization which they claim to be more suitable for spiking neural nets than the vanilla version. They show that their approach outperforms a few earlier spiking neural network approaches on the SLoClas dataset.

**Strengths:**

+ Spiking neural networks are an interesting and important research direction
 + Sound localization and classification are relevant perception problems
 + Models able to solve multiple tasks are a relevant research direction also in the auditory domain

**Weaknesses:**

1. Unclear why this particular combination of problems and approaches
 1. Claims not supported well by experiments and evidence
 1. Lack of strong baselines
 1. Very narrow evaluation on only a single, quite specific dataset
 1. Method is not really understandable from the paper, lots of details missing


Overall, I think this paper needs a major overhaul in terms of motivation, presentation and (most likely) additional experiments. I don't see how it could be accepted even if many of the questions could be answered in a discussion. I think it should be thoroughly revised and resubmitted, so reviewers can assess the revised version. To be honest, it should not have left the PIs desk like this. In an effort to be constructive, I will detail my concerns below, but I don't see much potential for a change in score.

**Questions:**

### 1. Why this combination of problems and approaches

It is not clear to me what the goal of the paper is. If it's about sound localization and classification, why do we need spiking neural nets? If it's about spiking neural nets, why such a narrow focus on this particular (combination of tasks)? What motivates the development of the LN layer for spiking nets? The paper reads like a very specific approach to a very specific problem, where the approach is not motivated by the requirements of the problem. Thus, it remains unclear to me what we can learn from this paper.


### 2. Claims not supported well by experiments and evidence

The paper makes four main claims at the end of the introduction. The first three of them are not supported by evidence in my opinion.

 1. The authors claim their paper is the first to use multi-task learning in SNNs and audio-related tasks. That may well be the case, by why is this a contribution? There is little evidence presented that multi-task learning is necessary to achieve the goal (sound localization and classification) or that it improved performance on either of the tasks. Table 3 shows that training each task individually works almost as well on localization and equally well on classification. As there are no error bars given, it remains difficult to judge whether there is an effect at all.

 1. The Spiking Hybrid Attention Fusion (SHAF) mechanism is presented as a contribution of the paper. However, I could not follow its description in the paper because too many symbols were not defined. Moreover, it is not clear to me whether this mechanism could actually be implemented with spiking neurons on neuromorphic hardware, since Q,K,V are claimed to be real-valued.

 1. The training-inference-decoupled LN method (DSLN) is presented as a contribution of the paper. Again, I could not follow what is happening since too many symbols were undefined. Generally, it is not clear to me what's the goal here. Since the whole point of layer norm is to normalize by activation statistics of other units in the same layer, I don't understand how the authors want to absorb it into the weights of the previous layer (seems to be the goal of Eq. 15) and why they drop the variance normalization (if getting rid of the sqrt is the goal, they could, e.g., use mean absolute deviation instead).



### 3. Lack of strong baselines

The authors claim to outperform earlier methods. That might be the case for the two spiking neural nets in Table 1, but how strong are these baselines? The ANN baseline casts some doubt: Why would the spiking version outperform an ANN baseline? What's the mechanism that makes a spiking net perform better than one that doesn't restrict itself to spiking? This seems to be an implausible claim or a very weak baseline.



### 4. Narrow evaluation

In case the goal is not to solve these two particular tasks (sound localization and classification) but to make a contribution to spiking neural nets in general, the paper would need a more thorough evaluation of a broader set of problems/datasets to demonstrate the usefulness of the method. If, however, the goal is to solve these particular two tasks, then I think there are better approaches than SNNs and it is not clear why the authors focus on SNNs.


### 5. Lack of clarity in methods

I found the description of the methods extremely hard to follow and could not resolve a number of questions. Part of the reason is that in many cases the motivation for doing something is not spelled out clearly at the outset, another part is that many symbols are simply not defined, not explained or their dimensions remain unclear. A few examples:

 1. Fig. 2: Meaning of N, F, B_g and B_m are unclear. They are not defined in the figure caption. N seems to be used at multiple places for multiple different things. In this Fig. it might refer to the number of microphone pairs, but later (Eq. 8) it shows up again with a different meaning. It looks like T x F x E from Fig. 2 might correspond to T x N x D in Eq. 8, but since neither are defined, I can only guess.

 1. The meaning of the symbols in Eq. 8 is unclear. First, the dimensions T x N x D: What does each mean? If T refers to time, does this mean attention extends over time? What are the dimensions of W_Q, W_K and W_V? and what does the product X_alpha W_Q mean? Why is there a batch norm around this product? That's not usually the case in Transformer attention. What is the meaning of SN(.)? The action potential symbols in Fig. 3 do not really help. What are the dimensions of SN(.), both input and output?

 1. Section 4.3, in particular Eq. 15 remains unclear to me. W, W'_*,* are only defined in terms of dimensions, but it's neither clear what they are, how they come about nor what m and n in their dimensions mean. Also, A_i has not been introduced.

---

> ### Author Response · Authors · 2023-11-22
> **Response to Reviewer zvoL (Part 1/3)**
>
> We appreciate your detailed comments. We would like to address your concerns and questions below.
>
> >Why this combination of problems and approaches?
>
> **R:** Our research primarily focuses on the domains of spiking neural networks and audio.
> The motivation of our work has been illustrated in Introduction section of our manuscript and we will give more discussion about the reason that we use SNNs in joint sound localization and classification.
>
> Firstly, as we know, SNN has the characteristics of bio-interpretability, lower power consumption, higher robustness, and better ability to extract spatio-temporal features. So it provides a more energy-efficient and bio-plausible way to perform such tasks in the audio field.
>
> Secondly, for humans, localizing and identifying sound sources simultaneously through binaural cues is a crucial and basic ability. Therefore, it’s very meaningful to investigate the information sharing ability of brain-inspired SNNs from the SELC task.
>
> Thirdly, previous work about SNNs has largely focused on the computer vision and the performance of SNN in handling other modal data has not been fully explored. Different from images, audio data has a temporal dimension and is more sparse, posing new challenges for SNN architectures. For example, how to integrate different acoustic features, how to normalize features at the frame level, and how to design low-power networks for audio.
>
> Finally, if we want to employ SNN in sound localization and classification, we have to solve the above basic problems in the specific task first. LN is a widely used normalization method to process temporal data like audio in ANNs. However, the vanilla LN do not conform to the calculation characteristics of SNNs, since it needs to be calculated dynamically during runtime and involves a substantial number of floating-point operations. Therefore, it’s very essential to develop a suitable LN method for SNNs.
>
> >Claims not supported well by experiments and evidence.
>
> **R:** **Multi-task learning.** Due to the powerful performance of our model and the initial high accuracy of sound source classification, multitask learning has limited impact on further improving sound source classification accuracy. Instead, it focuses more on enhancing sound source localization accuracy.
> Our results indicate that multitask learning is feasible in SNNs, which will also help promote the development of SNNs.
>
> **SHAF mechanism.** The SHAF mechanism consists of spike-form cross-attention and self-attention, which is able to integrate acoustic features. In SNNs, the direct input of LIF neurons can be real-valued [1]. The float-point $Q,K,V$ refers to the results of $XW$, serving as the input of LIF neurons. So it can be deployed on neuromorphic hardware.
>
> **DSLN method.** Our goal is to fold the LN calculation process into other layers like FC while retaining as many statistics as possible so that we can eliminate the time cost and computational energy consumption of DSLN. In practice, we choose to use a fixed variance and dynamically calculate the mean in the stage of inference. The re-parameterization procedure are derived in Eq. 15.
>
> Here, we conduct more ablation experiments on SHAF and DSLN to demonstrate the effectiveness of our method.
> For the SHAF, we apply SpikSLC-Net-2-128 and the model has two FC layers in each head. For the DSLN, to avoid the impact of head design on model performance, we apply SpikSLC-Net-3-128 with lightweight heads which are equipped with only one FC layer.
> The results are listed in Tab. R3-1 and Tab. R3-2. As we can see, their improvement in model performance is mainly reflected in the DoA estimation task, since the sound localization is more challenging than classification.
> The SCA and SSA modules in SHAF both can greatly improve the performance of the model. And the DSLN module can effectively alleviate overfitting in larger model.
>
> | Setting | DoA MAE(&deg;) | DoA Acc(\%) | SEC Acc(\%) |
> | :---- | :----: | :----: | :----: |
> | W/O SCA & SSA | 1.10 | 95.90 | 99.29 |
> | W/O SSA | 0.76 | 96.81 | 99.35 |
> | W/O SCA | 0.85 | 96.57 | 99.32 |
> | W/ SCA & SSA | **0.47** | **97.46** | 99.33 |
> **Table R3-1: Ablation studies on SHAF.**
>
> | Setting | DoA MAE(&deg;)  | DoA Acc(\%) | SEC Acc(\%) |
> | :---- | :----: | :----: | :----: |
> | W/O DSLN | 1.06 | 95.67 | 99.33 |
> | W/ DSLN | **0.53** | **97.32** | **99.45** |
> **Table R3-2: Ablation studies on DSLN.**

---

> ### Author Response · Authors · 2023-11-22
> **Response to Reviewer zvoL (Part 2/3)**
>
> >Lack of strong baselines.
>
> **R:** Our framework is based on full SNNs, so it’s reasonable to compare our model with other SOTA SNN methods on the same datasets they used. The previous best ANN method for sound localization and classification on SLoClas dataset is GCC-PHAT-CNN. In order to demonstrate the superiority of our model, we utilize a modified version of EINV2 [2] which is originally designed for sound event detection and implement an ANN version of our model (SLC-Net) for additional experiments. The corresponding results are shown in Tab. R3-3. The advantage of our model in DoA estimation is very obvious. According to the characteristics of SNN, we think that it is very possible for SNN to outperform ANN in specific cases. Additionally, it is plausible that the performance of the current best ANN model is still insufficient.
>
> | Method | DoA MAE(&deg;) | DoA Acc(\%) | SEC Acc(\%) |
> | :---- | :----: | :----: | :----: |
> | GCC-PHAT-CNN |4.88 | 84.50 | 78.90 |
> | EINV2 | 0.80 | 96.92 | 98.08 |
> | SLC-Net-2-128 | 0.87 | 96.52 | 99.11 |
> | MTPC-CSNN | 1.02 | 94.72 | - |
> | Hybrid Coding | 0.60 | 95.61 | - |
> | SpikSLC-Net-3-128 (Ours) | **0.45** | **97.39** | **99.42** |
> **Table R3-3: Comparison of different methods on SLoClas dataset.**
>
> >Narrow evaluation.
>
> **R:** Here, we conduct more experiments on different datasets and analyze the computation complexity and estimated power consumption to show the superiority of our SNN approach on two tasks.
> Since there are limited datasets and models for joint sound localization and classification, we construct a suitable dataset named DCASE 2019 Part. The dataset is the part of the DCASE 2019 dataset without overlapping sound sources. It has 11 sound sources and the azimuth angle interval is 10 degrees. When evaluating models on DCASE 2019 Part dataset, the threshold for accuracy calculation is set at 5 degrees, and it is set to 2.5 degrees on SLoClas dataset.
>
> In SNNs, the synaptic operations (SOPs) are used to describe the computation complexity, which is calculated by
>
> $\operatorname{SOPs}(l)=fr \times T \times \operatorname{FLOPs}(l)$,
>
> Where $l$ is a layer in SpikSLC-Net and $fr$ represents the firing rate of the input spike train, and $T$ denotes the time step. $\operatorname{FLOPs}(l)$ refers to floating point operations of $l$, which is the number of multiply-and-accumulate (MAC) operations. And SOPs is the number of spike-based accumulate (AC) operations.
> Refer to previous works [1][3], we assume that the operations are implemented on the 45nm hardware[4], where the theoretical energy consumption $EMAC = 4.6pJ$ and $EAC = 0.9pJ$.
>
> The results are shown in Tab. R3-4 and Tab. R3-5. The number of operations (OPs) refers to SOPs in SNN and FLOPs in ANN.
>
> As we can see, our models achieve the best performance while maintaining the lowest energy consumption compared with ANN models.
>
> | Method | DoA MAE(&deg;) | DoA Acc(\%) | SEC Acc(\%) |
> | :---- | :----: | :----: | :----: |
> | SLC-Net-2-128| 3.89 | 80.42 | 99.88 |
> | EINV2 |7.27 | 55.29 | 99.77 |
> | SpikSLC-Net-1-64 (Ours) | **2.42**| **86.08** | **1.00** |
> **Table R3-4: Comparison of different methods on DCASE 2019 Part dataset.**
>
> | Method | Param (M)|OPs (G) | Power (mJ) |
> | :---- | :----: | :----: | :----: |
> | GCC-PHAT-CNN | 4.17 | 0.048 | 0.143 |
> |EINV2 | 85.45 | 8.96 | 20.562 |
> |SLC-Net-2-128 | 4.15 | 0.067 | 0.152 |
> |SpikSLC-Net-1-128 (Ours) | **3.61** | **0.017** | **0.031** |
> | SpikSLC-Net-3-128 (Ours) | 4.68 | 0.056 | 0.066 |
> **Table R3-5: Comparison of the number of parameters, computation complexity and estimated power consumption.**

---

> ### Author Response · Authors · 2023-11-22
> **Response to Reviewer zvoL (Part 3/3)**
>
> >Lack of clarity in methods.
>
> **R:** Thanks for your carefully reading. We will align the symbols and add more detailed clarifications in the revised version.
>
> In Fig. 2, $N$, $F$, $B_g$ and $B_m$ represent the the total microphone pairs number, the frames number, the feature dimension of each frame in GCC-PHAT and the feature dimension of each frame in log-Mel spectrogram, respectively.
> In Eq. 8, the symbols are general, where $N$ is equal to $F$ in Fig. 2 and $D$ is equal to $E$ in Fig. 2.
>
> In the dimensions $T \times N \times D$, $T$ denotes the time step, $N$ denotes the frames number, and $D$ denotes the embedding dimension. The attention operation indeed extends over time step.
> The dimensions of $W_Q$, $W_K$ and $W_V$ are $D \times D$. The product $X_{\alpha} W_Q$ means projecting each frame of data at each time step to embedding space. According to [1], the batch norm around this product can normalize the embeddings and prevent neurons from firing dense spikes.
> $\operatorname{SN(.)}$ represents the LIF neuron layers and the dimensions of input and output are both $T \times N \times D$.
>
> In Eq. 15, $W$ represents the parameters of the previous Linear layer. $m$ and $n$ denote the input channel dimension and output channel dimension, respectively.  $A_i$ and $W'_i$ are the expressions of constants that result from previous calculations.
>
>
>
> Reference:
>
> [1] Zhaokun Zhou, Yuesheng Zhu, Chao He, Yaowei Wang, Shuicheng Yan, Yonghong Tian, and Li Yuan. Spikformer: When spiking neural network meets transformer. arXiv preprint arXiv:2209.15425, 2022.
>
> [2] Cao, Yin and Iqbal, Turab and Kong, Qiuqiang and An, Fengyan and Wang, Wenwu and Plumbley, Mark D. An improved event-independent network for polyphonic sound event localization and detection. ICASSP 2021-2021 IEEE International Conference on Acoustics, Speech and Signal Processing (ICASSP).
>
> [3] Man Yao, Guangshe Zhao, Hengyu Zhang, Yifan Hu, Lei Deng, Yonghong Tian, Bo Xu, and Guoqi Li. Attention spiking neural networks. arXiv preprint arXiv:2209.13929, 2022.
>
> [4] Mark Horowitz. 1.1 computing’s energy problem (and what we can do about it). In 2014 IEEE International Solid-State Circuits Conference Digest of Technical Papers (ISSCC), pages 10–14. IEEE, 2014.

---

### Official Review · Reviewer_zsh4 · 2023-10-31

**Soundness:** 3 good
**Presentation:** 3 good
**Contribution:** 4 excellent
**Rating:** 8
**Confidence:** 2

**Summary:**

The paper presents a Spiking Neural Network based framework for sound source localization and classification. The framework incorporates a novel Spiking Hybrid Attention Fusion (SHAF) mechanism and a unique training-inference-decoupled Layer Normalization method (DSLN), and achieves state-of-the-art performance on the SLoClas dataset with minimal computational steps.

Overall, within the spike neural network framework, this is a useful contribution. Although I have doubts about it's performance relative to traditional ANNs and CNNs, it is nice to see SNNs applied to a wider array of complex audio tasks.

**Strengths:**

The paper introduces a novel approach to simultaneous sound source localization and classification using SNNs. I haven't seen such use of SNNs for audio-related multi-task learning

The introduction of the Spiking Hybrid Attention Fusion mechanism, is interesting. This mechanism seems to capture temporal dependencies and aligns relationships among diverse features.

Layer Normalization: The DSLN method proposed for SNNs addresses the challenges associated with dynamic calculation during runtime in vanilla layer normalization. This method reduces the floating-point operations required, making it more suitable for SNNs.

Energy Efficiency: The framework’s design is motivated by energy efficiency, which is a significant consideration for deploying models in real-world applications, especially on edge devices.

The paper shows strong  performance on the SLoClas dataset

The ablation studies are good and show that the proposed layer norm maintains strong performance while reducing the overall computation

**Weaknesses:**

Baselines - The method compares with other SNN baselines and one ANN baseline. However I would like to see comparisons with other recent ANN methods. For example the method in (https://arxiv.org/pdf/2010.06007.pdf) is able to localize and separate speech to 2.1 degrees, although it uses 8 microphones to do so.

Achieving 99% accuracy with only 2 timesteps during inference is impressive, but it does raise questions about overfitting. It would strengthen the paper to have some real world examples or examples on a different dataset besides the Sloc dataset.

**Questions:**

Do you have any other results to compare against traditional ANN methods?

---

> ### Author Response · Authors · 2023-11-22
> **Response to Reviewer zsh4**
>
> Thank you for your recognition of our work. And now we will address your concerns in the following.
>
> >Do you have any other results to compare against traditional ANN methods?
>
> **R:** Since there are limited datasets and models for joint sound localization and classification, we construct a suitable dataset named DCASE 2019 Part. The dataset is the part of the DCASE 2019 dataset without overlapping sound sources. It has 11 sound sources and the azimuth angle interval is 10 degrees.
> For traditional ANN methods, we employ a modified version of EINV2 [1] which is originally used in sound event localization and detection and implement an ANN version of our model (SLC-Net). When evaluating models on DCASE 2019 Part dataset, the threshold for accuracy calculation is set at 5 degrees, and it is set to 2.5 degrees on SLoClas dataset.
> The results are shown in Tab. R2-1. As we can see, our method outperforms the ANN methods on both sound event localization and classification tasks.
>
> | Method | Dataset | DoA MAE(&deg;) | DoA Acc(\%) | SEC Acc(\%) |
> | :---- | :---- | :----: | :----: | :----: |
> | GCC-PHAT-CNN | SLoClas |4.88 | 84.50 | 78.90 |
> | EINV2 | SLoClas |0.80 | 96.92 | 98.08 |
> | SLC-Net-2-128 | SLoClas | 0.87 | 96.52 | 99.11 |
> | SpikSLC-Net-3-128 (Ours) | SLoClas | **0.45** | **97.39** | **99.42** |
> | SLC-Net-2-128 | DCASE2019 Part | 3.89 | 80.42 | 99.88 |
> | EINV2 | DCASE2019 Part |7.27 | 55.29 | 99.77 |
> | SpikSLC-Net-1-64 (Ours) | DCASE2019 Part | **2.42**| **86.08** | **1.00** |
> **Table R2-1: Comparison of different methods.**
>
> Reference:
>
> [1] Cao, Yin and Iqbal, Turab and Kong, Qiuqiang and An, Fengyan and Wang, Wenwu and Plumbley, Mark D. An improved event-independent network for polyphonic sound event localization and detection. ICASSP 2021-2021 IEEE International Conference on Acoustics, Speech and Signal Processing (ICASSP).

---

### Official Review · Reviewer_VDuB · 2023-11-06

**Soundness:** 2 fair
**Presentation:** 3 good
**Contribution:** 3 good
**Rating:** 6
**Confidence:** 3

**Summary:**

This paper designed a multi-task spiking neural network (SNN), by incorporating spiking self-attention modules and cross-attention modules that can capture temporal dependencies, to solve both sound source localization and classification tasks. The authors further introduced training-inference-decoupled layer normalization (DSLN), and demonstrated similar performance in benchmarking on SEC tasks with fewer time steps.

**Strengths:**

1. This paper showed novelty as a first implementation in multi-task learning with full SNNs in audio domain, while previous papers on SNNs have focused more on single classification task.
2. Strong benchmarking performance in showing superior accuracy in both SEC and localization tasks, lower MAE for localization task, at fewer time steps. , which further brought possibility on short latency given few time steps required by SNN.

**Weaknesses:**

1. Most previous SEC task reported F1 score to account for potential bias in sound classes. Would it be possible to show F1 score benchmarking with previous SEC task (potentially could leverage some DCASE datasets) instead? Such high accuracy was unclear to me if there were any biases, and unclear whether your algorithm has specific bias on precision vs. recall.
2. Ablation studies of DSLN do not suggest the difference is statistically significant, suggesting a relatively small or even no impact on the strong performance in SEC and localization tasks.
3. Additionally, ablation studies did not show how self-attention and cross-attention modules could play an essential role in decoding these tasks. The non-significant differences in numbers of SHAF blocks and embedding dimensions (Table 2) posed a confusing and open question on where the major performance benefits from. It would be more persuading by providing a stronger ablation study, showing how SHAF blocks contribute to the essential performance boosting here, while the rest of modules (multi-task) remained in the model.

**Questions:**

1. Following above in weaknesses, I think it would be helpful to perform ablation studies, showing the role of the proposed SHAF blocks and DSLN specifically in the superior performance. The fact that ablation studies did not show strong impact wrt different hyperparameters in SHAF and LN modules, pose a question on whether the major benefit comes from the multi-task heads, or different algorithmic complexity of this SNN vs. previous SNN.
2. Can authors provide benchmarking with F1 score, precision, recall?
3. Can authors provide an algorithmic complexity analysis of this SNN vs. previous SNN?

---

> ### Author Response · Authors · 2023-11-22
> **Response to Reviewer VDuB**
>
> Thank you for your constructive comments and suggestions for improvement. We would like to address your concerns and answer your questions in the following.
>
> >Following above in weaknesses, I think it would be helpful to perform ablation studies, showing the role of the proposed SHAF blocks and DSLN specifically in the superior performance.
>
> **R:** We perform more ablation studies on SHAF and DSLN modules to show their significance in our models.
> For the SHAF experiment, we apply SpikSLC-Net-2-128 and the model has two FC layers in each head. For the DSLN experiment, to avoid the impact of head design on model performance, we apply lightweight heads which are equipped with only one FC layer.
> The results are listed in Tab. R1-1 and Tab. R1-2. As we can see, their improvement in model performance is mainly reflected in the DoA estimation task, since the sound localization is more challenging than classification.
> The SCA and SSA modules in SHAF both can greatly improve the performance of the model. Besides, the DSLN module can effectively alleviate overfitting in larger model.
>
> | Setting | DoA MAE(&deg;) | DoA Acc(\%) | SEC Acc(\%) |
> | :---- | :----: | :----: | :----: |
> | W/O SCA & SSA | 1.10 | 95.90 | 99.29 |
> | W/O SSA | 0.76 | 96.81 | 99.35 |
> | W/O SCA | 0.85 | 96.57 | 99.32 |
> | W/ SCA & SSA | **0.47** | **97.46** | 99.33 |
> **Table R1-1: Ablation studies on SHAF.**
>
> | Setting | Architecture | DoA MAE(&deg;)  | DoA Acc(\%) | SEC Acc(\%) |
> | :---- | :----:  | :----: | :----: | :----: |
> | W/O DSLN | SpikSLC-Net-1-128 | 0.83 | 97.18 | 99.28 |
> | W DSLN | SpikSLC-Net-1-128 | **0.55** | **97.56** | 99.32 |
> | W/O DSLN | SpikSLC-Net-3-128 | 1.06 | 95.67 | 99.33 |
> | W/ DSLN | SpikSLC-Net-3-128 | **0.53** | **97.32** | **99.45** |
> **Table R1-2: Ablation studies on DSLN.**
>
> >Can authors provide benchmarking with F1 score, precision, recall?
>
> **R:** In order to fully demonstrate the superiority of our model, we select the part of the DCASE 2019 dataset without overlapping sound sources for additional experiments (DCASE 2019 Part). It has 11 sound sources. What’s more, we implement an ANN version of our model (SLC-Net) for comparison on SLoClas dataset and DCASE 2019 Part dataset. The corresponding metrics are shown in Tab. R1-3. For DCASE 2019 Part, the threshold for accuracy calculation is set at 5 degrees, as the azimuth angle interval is 10 degrees. We can find that our method displays absence of potential bias in sound classes, ensuring unbiased precision and recall measurements.
>
> | Method | Dataset | DoA MAE(&deg;)  | DoA Acc(\%) | SEC Acc(\%) | Precision | Recall | F1 |
> | :---- | :---- | :----: | :----: | :----: | :----: | :----: | :----: |
> | GCC-PHAT-CNN | SLoClas |4.88 | 84.50 | 78.90 | 0.9022 | 0.7890 | 0.7956 |
> | SLC-Net-2-128 | SLoClas | 0.87 | 96.52 | 99.11 | 0.9912 | 0.9910 | 0.9911 |
> | SpikSLC-Net-1-128 (Ours) | SLoClas | 0.51 | 97.53 | 99.30 | 0.9931 | 0.9930 |0.9930 |
> | SpikSLC-Net-2-128 (Ours)  | SLoClas | 0.47 | 97.46 | 99.33 | 0.9934 | 0.9933 | 0.9933 |
> | SpikSLC-Net-3-128 (Ours)  | SLoClas | 0.45 | 97.39 | 99.42 | 0.9942 | 0.9943 | 0.9942 |
> | SLC-Net-2-128 | DCASE2019 Part | 3.89 | 80.42 | 99.88 | 0.9988 | 0.9988 | 0.9988 |
> | SpikSLC-Net-1-64 (Ours) | DCASE2019 Part | 2.42| 86.08 | 1.0 | 1.0 | 1.0 | 1.0 |
> **Table R1-3: Comparison of different metrics.**
>
> >Can authors provide an algorithmic complexity analysis of this SNN vs. previous SNN?
>
> **R:** Since previous SNN methods do not release their code and their contributions focus on coding methods, it is difficult to make a fair comparison of algorithm complexity. Moreover, it is worth noting that other methods are unable to perform both sound source localization and recognition simultaneously, as we do. Here, we present the algorithmic complexity analysis of our models and other ANN models on SLoClas dataset in Tab. R1-4. The number of operations (OPs) refers to synaptic operations (SOPs) in SNN and FLOPs in ANN. It can be seen that our method achieves high accuracy while requiring the lowest algorithmic complexity.
>
> | Method | OPs (M) |
> | :---- | :----: |
> | GCC-PHAT-CNN | 48.08 |
> |SLC-Net-2-128 | 66.82 |
> |SpikSLC-Net-1-128 (Ours) | **17.15** |
> | SpikSLC-Net-3-128 (Ours) | 55.80 |
> **Table R1-4: Comparison of algorithmic complexity.**

---

### Author Response · Authors · 2023-11-23
**To All Reviewers**

We thank all reviewers for their time, patience, and constructive comments to help us improve our paper. Based on the helpful comments of the reviewers, we have revised the paper carefully. The main revisions in this update include symbols alignment and additional results on other ANNs and datasets.

---

### Meta-Review · Area_Chair_eYJU · 2023-12-06

**Metareview:**

The authors propose an algorithm for sound source localization and classification using spiking neural networks (SNNs). It uses a spiking hybrid attention fusion (SHAF) mechanism that uses spiking self-attention and spiking cross-attention modules. A training-inference-decoupled layer norm (DSLN) method is also proposed for SNNs to improve efficiency. The model is trained on 2 tasks – direction-of-arrival estimation and event classification.

Novelty lies in the specific combination of the use of SNNs for the two tasks of localization and event classification. There is some novelty in the modules for building such a model, specifically spiking cross attention (SCN) and DSLN. That said, SCN is an extension of the previously proposed spiking self attention block, extended as one would do cross-attention in typical transformers (Q from one source, KV from the other).

The authors claim that LayerNorm is very expensive in the context of SNN. LN being a standard component in almost all modern transformer models, so the impact is arguably small especially since DSLN performs similarly as LN. The authors provide quantitative results in their rebuttal to show speed ups of the overall network, but not specifically DSLN, so it is unclear how significant the differences are. Performance-wise, it’s quite similar.

There are not many works using the SloClas dataset. Therefore, there are some concerns on how strong the baselines and gains are. Perhaps the authors should consider a more widely used dataset for event classification to show what the model is capable of.  Furthermore, it is unclear why the two tasks have to be learned jointly, as pointed out by one of the reviewers. It would have helped if the authors considered showing the utility of the proposed models on a more widely studied task with larger training sets (potentially using multiple feature representations to show benefits of SCA). The current combination of SNN + sound localization + even classification is too narrow, making it harder to extrapolate the utility for tasks with larger datasets and strong baselines.

SloClas is a small dataset (~23 hours of audio, 10 sound classes), so it is unclear whether the proposed method would generalize to tasks with larger datasets. Reviewers recommended using DCASE. The authors provided results using a subset of DCASE set (DCASE 2019 Part), but this will likely need a more thorough review. But again, this is a smaller task and would be much better if a more widely studied task with larger datasets is considered.

Reviewers pointed out that the ablations do not necessarily show the benefits of the proposed modules since the difference in quality is small. The authors use different size models in their rebuttal to show the effect, but this may not be a fair comparison given larger models work better.

**Justification For Why Not Higher Score:**

As mentioned above, the presented work is evaluated in a narrow setting making it difficult to evaluate the general applicability of the ideas on other tasks. The proposed components also have limited novelty compared to prior works, and they have not been satisfactorily evaluated.

**Justification For Why Not Lower Score:**

N/A

---

### Decision · Program_Chairs · 2024-01-16

Reject